# Ibrutinib in Combination with Lenalidomide Revlimid/Dexamethasone in Relapsed/Refractory Multiple Myeloma (AFT-15)

**DOI:** 10.3390/cancers17152433

**Published:** 2025-07-23

**Authors:** Yvonne Efebera, Vera Suman, Shira Dinner, Taylor O’Donnell, Ashley Rosko, John Mckay, Peter Barth, Patrick Hagen, Saad Usmani, Paul Richardson, Jacob Laubach

**Affiliations:** 1MPH, OhioHealth, Columbus, OH 43214, USA, yvonne.efebera@ohiohealth.com; 2Alliance Foundation Trials Statistics and Data Center, Mayo Clinic, Rochester, MN 55905, USA; suman@mayo.edu; 3Northwestern Medical Center, Division of Hematology, Chicago, IL 60611, USA; shira.dinner@nm.org; 4Alliance Foundation Trials, LLC, Boston, MA 02115, USA; todonnell@alliancefoundationtrials.org; 5Division of Hematology, The Ohio State University, Columbus, OH 43210, USA; ashley.rosko@osumc.edu; 6Wake Forest Baptist Health, Winston-Salem, NC 27157, USA; jtmckay@wakehealth.edu; 7Lifespan Cancer Institute, Providence, RI 02906, USA; peter.barth@lifespan.org; 8Loyola University Medical Center, Maywood, IL 60153, USA; patrick.hagen@lumc.edu; 9Memorial Sloan Kettering Cancer Center, New York, NY 10065, USA; usmanis@mskcc.org; 10 Dana-Farber/Partners Cancer Care, Boston, MA 02115, USA; paul_richardson@dfci.harvard.edu (P.R.); jacobp_laubach@dfci.harvard.edu (J.L.)

**Keywords:** multiple myeloma, ibrutinib, lenalidomide, relapse and refractory multiple myeloma

## Abstract

We conducted a 3 + 3 phase Ib trial to determine the tolerability and efficacy of Ibrutinib (IBR) in combination with lenalidomide (LEN) + dexamethasone (DEX) in patients with relapsed/refractory (RR) MM with at least one prior line of therapy. Thirteen patients were eligible for disease response. The median number of prior lines of therapy was 3 (2–9). Ten patients (76.9%) had an autologous stem cell transplant. All were exposed to an immunomodulatory drug (IMID; LEN [92.3%], pomalidomide [30.8%] or thalidomide [7.7%]) and a proteasome inhibitor (bortezomib [100%], carfilzomib [38.5%] or ixazomib [15.4%]). In total, 53.8% of patients were exposed to an anti-CD 38, and 7.7% were exposed to chimeric antigen receptor T cell therapy. The combination was well tolerated, with clinical activity seen. The overall response rate (partial response and better) was 15.4%. One patient maintained stable disease (SD) for 4.6 years. Five patients maintained SD for ≥2 cycles. The clinical benefit response (SD and better) was 61.5%. The all-oral combination of LEN with IBR was feasible and well tolerated, with evidence of clinical activity. Future directions for an all-oral combination with IBR and the accompanying real-world advantages include next-generation IMiDs such as pomalidomide, and the potent oral degraders in the CELMod category, specifically iberdomide and mezigdomide.

## 1. Introduction

Multiple myeloma (MM) is the second most common hematological malignancy. Prior to the availability of carfilzomib and pomalidomide, MM patients refractory to both lenalidomide (LEN) and bortezomib had an event-free survival of approximately 5 months and an estimated median overall survival (OS) of 9 months [1]. The median (OS) has been extended with the use of pomalidomide and carfilzomib to over 15 months [2]. However, all MM patients will eventually become resistant, and hence new combination treatments are needed urgently.

LEN, a thalidomide analog, is an immunomodulatory agent with antiangiogenic properties. Its mechanism of action remains to be fully characterized. LEN has been shown to inhibit the secretion of pro-inflammatory cytokines and increase the secretion of anti-inflammatory cytokines from peripheral blood mononuclear cells [3]. LEN-based therapy is considered a standard of care in the settings of newly diagnosed, maintenance, and relapsed MM [4].

Ibrutinib (IBR) is a Bruton’s tyrosine kinase (Btk) inhibitor. Btk is a member of the Tec family of tyrosine kinases, the activation of which regulates B-cell development and plays an important role in antibody production; it has become a therapeutic target in MM [5,6]. In vitro studies have demonstrated that treatment of myeloma cells with IBR results in a modest reduction in cell viability and the induction of apoptosis. IBR has been shown to directly inhibit osteoclastic bone resorption and migration of SDF-1-induced MM cells, as well as to prevent the release of multiple osteoclast-derived tumor growth factors and the adhesion of MM cells to bone marrow stromal cells (BMSCs) [7,8,9]. IBR has also been shown to inhibit BTK-driven NK-kB p65 activity to overcome bortezomib resistance [5,8]. The safety and efficacy of IBR in combination with DEX [10], LEN [11], bortezomib/DEX [12], and carfilzomib/DEX [13] have been demonstrated in patients with RR MM.

The combination of IBR and LEN in patients with RR non-Hodgkin’s lymphoma and chronic lymphocytic leukemia [14,15,16,17,18,19] demonstrated promising activity, with neutropenia and infections being the most common adverse effects.

Preclinical research in MM suggests a synergism between LEN and IBR in MM [8]. Both IBR and LEN downregulate a master transcriptional factor, IRF4, that mediates myeloma cell survival, and disrupts bone marrow stromal support. Both IBR and LEN exert effects on the bone marrow microenvironment with overlapping but distinct targets that influence MM cell growth and survival. The combination of LEN and IBR is therefore an attractive therapeutic strategy, not only as a potent anti-MM regimen but also in terms of positively impacting bone disease.

## 2. Study Design and Treatment

A 3 + 3 phase I clinical trial involving dose escalation was conducted to examine the safety, tolerability, and antitumor activity of IBR in combination with LEN and DEX in individuals aged 18 years or older with symptomatic malignant myeloma who had undergone at least 2 prior lines of therapy. Due to changes in standard of care during the study, eligibility was modified in May 2022 to allow enrollment of patients with 1 or more prior lines of therapy. A total accrual of 19–25 patients was planned. The study opened May 2019 and closed May 2023 due to slow accrual, and hence dose expansion was not performed.

Within 14 days of registration, potential participants underwent the following evaluations to determine eligibility: a physical exam with a review of medical history and symptoms, an assessment of Eastern Cooperative Oncology Group (ECOG) performance status, determination of hematologic and blood chemistries, determination of TSH, LDH, and uric acid levels, electrocardiographic testing, bone marrow biopsy, complete-body CT, and assessment of serum immunoglobulins, monoclonal protein and free light chain and beta 2 microglobulin levels. Eligible patients received their assigned dose level at the time of registration.

Eligibility criteria included measurable disease (defined as at least two of the following: serum M-protein > 0.5 g/dL, 24-hour urine M-protein > 200 mg, and involved serum-free light chain (FLC) level > 10 mg/dL (>100 mg/L), provided the serum FLC ratio is abnormal); an ECOG performance score of 0–1; experience of at least 1 prior line of therapy with demonstrated disease progression after the most recent line of treatment; disease progression within 60 days of completion of the last treatment regimen or failure to achieve minimal response while on the last treatment (Per International Myeloma Working Group [IMWG]); adequate blood chemistry (Appendix A); a negative serum pregnancy test within 7 days prior to registration, if premenopausal; and agreement to comply with measures to prevent pregnancy for at least 90 days after the last dose of LEN.

Exclusion criteria included progression on LEN at a dose of more than 10 mg; prior treatment with IBR or any other protein kinase-inhibitory drug or drug targeting the B cell receptor signal transduction pathway; treatment with alkylators, anthracyclines, or radiation therapy within 21 days prior to registration, exposure to monoclonal antibody within 6 weeks prior to registration; treatment with high-dose corticosteroids, immune-modulatory drugs or proteasome inhibitors within 14 days prior to registration; clinically significant cardiovascular disease, or NYHA Class 3 or 4 cardiac disease; myocardial infarction, stroke or intracranial hemorrhage within 6 months prior to registration; major surgery within 4 weeks prior to registration; peripheral neuropathy grade > 2 within 14 days prior to registration; uncontrolled diabetes mellitus; uncontrolled, active systemic fungal, bacterial, viral, or other infections; antibiotic treatment of infection within 14 days prior to registration; known active infection with human immunodeficiency virus and hepatitis C virus or hepatitis B virus or any uncontrolled active systemic infection; exposure to strong cytochrome P450 (CYP3A) inhibitor within 7 days of registration; vaccination with live, attenuated vaccines within 4 weeks prior to registration; and pregnancy or breastfeeding. Enrollment was open to patients who had received daratumumab and/or an allogeneic stem cell transplant or had a history of another malignancy that was treated with curative intent and no active disease in the 3 years prior to registration.

This study was opened for enrollment after approval by the participating institutions’ Institutional Review Boards in accordance with assurances filed with, and approved by, the Department of Health and Human Services. The Alliance Foundation Trials Data and Safety Monitoring Board monitored the study twice a year to ensure safety and progress toward completion. All patients provided written informed consent. This study has been registered at Clinicaltrials.gov as NCT03702725.

### 2.1. Treatment

Three dose levels (DLs) were planned. The cycle length was 28 days. IBR was administered orally at a dose of 560 mg daily on DL1-2 and 840 mg on DL3. LEN was administered orally on days 1–21 at 15 mg for DL1 and at 25 mg for DL2-3. DEX was administered orally on days 1, 8, 15, and 22 at 40 mg if the patient’s age was <75 years or at 20 mg if their age was ≥75 years for DL1-3. Patients with a glomerular filtration rate ≥ 30 and <60 were treated in accordance with the manufacturer’s instructions with LEN dosed at 10 mg/day for all dose levels. No intra-patient dose escalation was permitted. Dose modification guidelines are provided in Appendix A.

### 2.2. Study Procedures

Prior to the administration of day 1 treatment for all treatment cycles and upon the discontinuation of all protocol treatments, patients underwent a physical exam, an assessment of performance status, and determination of hematologic and blood chemistries, LDH, uric acid levels, and toxicity (using CTCAE v5.0). Serum immunoglobulin, monoclonal protein and free light chain levels were assessed prior to each treatment cycle and upon the treatment’s discontinuation. An electrocardiogram was performed prior to and at the end of treatment. Upon completion of each treatment cycle, disease response was assessed using the IMWG Uniform Response Criteria. If a patient had extramedullary disease, seen on the baseline scan, then CT/MRI and/or PET/CT was to be performed upon the completion of cycle 6 or to confirm CR. A new cycle of treatment could begin if the ANC was ≥1000/μL and the platelet count was ≥50,000/μL; if any IBR- or LEN-related allergic reaction/hypersensitivity or sinus bradycardia/ other cardiac arrhythmia adverse event (AE) that occurred resolved to ≤grade 1; if any other IBR- or LEN-related AE that occurred resolved to ≤grade 2; and if all laboratory values (excluding Child Pugh score) met eligibility criteria levels. Patients were treated until disease progression, intolerability, or patient request to discontinue their participation..

### 2.3. Statistical Design and Methods

A 3 + 3 phase I clinical trial design was chosen to ascertain the maximum tolerate dose (MTD) of IBR in combination with LEN and DEX. Dose-limiting toxicities (DLTs) included the following events occurring during the first cycle of treatment: grade 4 thrombocytopenia; grade 4 neutropenia lasting more than 5 days or febrile neutropenia; grade 3 thrombocytopenia with bleeding or platelet transfusion; grade 3–4 hyperglycemia or a thrombotic/embolic event; or grade 3–4 non-hematologic toxicity (except grade 3 nausea, diarrhea, and vomiting), that were probably, possibly or definitely related to treatment as well as treatment delays ≥ 21 days for toxicity and treatment-related death. The dose escalation schema was as follows: (a) If none of the first 3 patients enrolled at a given dose level developed a DLT, then 3 patients would be enrolled at the next higher DL; if all the DLs had been exhausted, then an additional 3 patients would be enrolled at the current DL to establish it as the MTD. (b) If 2 or more of the first 3 patients enrolled at a given DL developed a DLT and only 3 patients had been enrolled for the next lower dose level, then an additional 3 patients would be enrolled at the next lower DL. Otherwise, enrollment would cease. (c) If 1 of the first 3 patients enrolled at a given DL developed a DLT, then 3 additional patients would be enrolled at the current DL. If 2 or more of the 6 patients treated at the current dose developed a DLT, and only 3 patients had been enrolled on the next lower dose level, then an additional 3 patients would be enrolled at the next lower DL. Otherwise, enrollment would cease. If 1 of the 6 patients treated at the current dose developed a DLT, then the next 3 cohorts of patients would be enrolled at the next higher DL. If all the DLs had been exhausted, then the current DL would be the MTD.

All patients who provided written informed consent and began protocol-directed therapy were included in the analysis of the safety and clinical outcome data.

The primary aim was to determine the MTD of IBR in combination with LEN and DEX. Secondary endpoints included the safety profile of IBR/LEN/DEX and the overall response rate. The safety profile was examined first, considering all toxicities regardless of attribution. The maximum grade of each toxicity a patient developed over the course of treatment was determined and then the percentage of patients developing that toxicity by grade was ascertained. The process was repeated for toxicities at least possibly related to treatment. The overall response rate (ORR) was defined as the percentage of patients who whose disease met the IMWG criteria for partial response, very good partial response or complete response (PR, VGPR, or CR) on two consecutive evaluations at least 4 weeks apart among all eligible patients who began protocol treatment [20,21]. The clinical benefit rate (CBR) was defined as the percentage of patients whose disease met the IMWG criteria for SD, MR, PR, VGPR, or CR on two consecutive evaluations at least 4 weeks apart among all eligible patients who began the protocol treatment.

A 95% binomial confidence interval was constructed for the ORR and CBR.

Progression-free survival time (PFS) was defined as the time from registration to the documentation of disease progression or death due to any cause. OS was defined as the time from registration to death due to any cause. The distributions of these times until the occurrence of events were estimated using the Kaplan–Meier method. The censoring scheme for PFS was such that patients who were alive without disease progression or died without disease progression more than 90 days after the last disease evaluation were censored at the first of the following events: their last disease evaluation or the start of a non-protocol treatment. Patients who were still alive at the point of last contact were censored when estimating the OS distribution.

Data quality was ensured via a review of the data by the Alliance Foundation Trials Statistics and Data Center and by the study chairperson following Alliance policies. All analyses were carried out using SAS 9.4M7 based on the study database, which was frozen; data for this report were downloaded and saved on 27 September 2024

## 3. Results

### 3.1. Patient Characteristics

From 1 March 2019 to 10 May 2023, 14 patients (DL1—6 pts; DL2—3 pts; DL3—5 pts) were enrolled prior to the trial closing due to slow accrual. One patient who was enrolled onto DL3 discontinued their participation prior to the start of treatment and was excluded from these analysis. Patient baseline characteristics of the remaining 13 patients are in Table 1. The median age was 64 years (45–84). The median number of prior lines of therapy was 3 (range: 2–9). Ten patients (76.9%) had a prior autologous stem cell transplant. All 13 patients were exposed and refractory to an immunomodulatory drug (IMID), namely LEN [92.3%], pomalidomide [30.8%] or thalidomide [7.7%]), and to a proteasome inhibitor (namely bortezomib [100%], carfilzomib [38.5%] or ixazomib [15.4%]). In addition, 53.8% of patients had been exposed to an anti-CD 38, and 7.7% had been exposed to chimeric antigen receptor T cell (CAR-T) therapy.

### 3.2. Dose Escalation

One of the first one patients enrolled onto DL1 developed a DLT, namely grade 3 non-viral hepatitis. Neither of the next three patients enrolled onto DL1 nor the three patients enrolled onto DL2 developed a DLT. Of the four patients who enrolled onto DL3, two were excluded from DLT determination having discontinued cycle 1 treatment; one patient was excluded due to COVID-19 infection and the other received 280 mg/day of IBR instead of the assigned 840 mg/day dose during cycle 1. The remaining two patients did not develop a DLT.

### 3.3. Treatment Course, Safety and Toxicity

All patients have now discontinued the protocol treatment. The median number of treatment cycles administered was 4 (range: 1–56). One patient enrolled onto DL1 received reduced doses of all three agents after cycle 1 due to grade 2 muscle weakness and fatigue. No other IBR reductions occurred.

Grade 2–5 toxicities reported across all cycles of treatment regardless of attribution are presented in Table 2. The patient on DL1 who developed grade 2 muscle weakness and fatigue after one cycle of treatment was later hospitalized for a soft tissue infection and cellulitis of the lower extremity that had spread and subsequently led to sepsis syndrome and death in the setting of progressive disease. This was considered unlikely to be related to IBR and LEN but possibly related to DEX. Another nine (69.2%) patients developed a grade 3–4 AE. The most common grade 3–4 toxicities reported were decreased lymphocyte count (grade 3: 30.8%; grade 4: 7.7%) and anemia (grade 3: 15.4%).

The median time on treatment was 3.1 months (range: 1.7–54.6 months). Treatment was discontinued in 13 patients, due to disease progression (7); AEs including viral infection (1) and delays of more than 4 weeks in treatment due to urinary tract infection with fatigue and diarrhea (1); patient refusal (1); death due to sepsis (1); a desire to proceed to CAR-T therapy (1); and other complicating medical conditions (recto-vaginal fistula [1]).

### 3.4. Clinical Outcome: Response and Survival

The distribution of objective responses among these 13 patients was as follows: one CR, one VGPR, nine cases of stable disease (SD), 1 PROG, and 1 NE. The patient with a CR was enrolled onto DL3; CR was documented after the first and second treatment cycles, but then the patient discontinued treatment due to a recto-vaginal fistula. One patient on DL1 showed an increase in response to VGPR (on two evaluations) and then showed disease progression 7.1 months post-registration. One patient on DL1 remained on the protocol treatment for 4.6 years with SD before discontinuing to proceed to CAR-T therapy. An additional five patients remained on the protocol treatment with SD for ≥2 consecutive cycles (Figure 1). Thus, the ORR was 15.4% (90% CI: 2.8–41.0%) and the CBR was 61.5% (90% CI: 35.5–83.4%).

As of September 2024, three patients were alive without disease progression, three patients were alive with disease progression, and seven patients had died. Causes of death were reported as multiple myeloma (2), gastrointestinal bleeding (1), neutropenia (1), sepsis syndrome (1), and unknown (2).

The median length of follow-up among the six patients who were still alive was 2.3 years (range: 2.0–4.5 years). The median PFS was 3.5 months (95%CI: 56 days–NE), with a 6-month PFS rate of 37% (95% CI: 17.1–80.2%) (Figure 2a). The median OS was 1.1 years (95%CI: 6.8 months–NE), with a 1 year OS rate of 53.8% (95% CI: 32.6–89.1%) (Figure 2b).

## 4. Discussion

We found that the all-oral combination of LEN with IBR was feasible and generally well tolerated, with evidence of clinical activity, including one CR, one VGPR and another six patients achieving sustained SD and remaining on treatment for three or more cycles. The MTD could not be established, and dose expansion could not be performed given the trial’s early closure. Only one DLT was seen at DL1.

Prior studies with the combination of IBR with DEX or with LEN, bortezomib or carfilzomib in RR MM have shown promising results. A phase 1 study of IBR plus LEN/DEX [11], where the doses of IBR tested were 420, 560, 700 and 840 mg daily with 25 mg of LEN daily for 21 of 28 days and DEX 40 mg weekly, showed an ORR of 7% and a CBR of 80% in the 15 patients treated. The MTD of IBR was 840 mg, with one DLT at this level and no DLT at other levels. There were no major differences in patient selection and outcomes in this study compared to ours. Our study, therefore, supports that the combination of IBR and an immunomodulator is feasible.

A phase 2 study examined four different regimens of IBR plus DEX in patients having undergone two or more prior lines of therapies [10]. The most promising combination was IBR at a dose of 840 mg daily with DEX dose of 40 mg weekly, with an ORR of 5%, and the percentage of patients with MR or a better outcome was 28%. The most common grade 3–4 AEs were anemia (9%) and thrombocytopenia (9%).

Enrollment to a phase 2 study on Ibrutinib 840 mg daily with bortezomib 1.3 mg/m^2^ on days 1, 4, 8, and 11 and with DEX 20 mg on days 1, 2, 4, 5, 8, 9, 11, and 12 of each 21-day cycle in patients with 1–3 prior lines of treatment was suspended due to serious (42%) and fatal (11%) infection rates reaching 50%. Grade 3/4 AEs occurred in 73% of patients, with fatal AEs in 15%. The ORR was 57%, with a median duration of response of 9.5 months [12].

The CBR was 78%, and the median duration of a clinical benefit response was 6.5 months.

A phase 2 study of IBR 840 mg daily with carfilzomib and DEX showed an overall best response rate of 71%, with a median duration of response of 6.5 months (range: 0.5–44.3). The most common grade 3–4 treatment-emergent AEs reported were anemia (17%), thrombocytopenia (17%) and hypertension (19%) [13].

To date, no targeted tyrosine kinase inhibitor has been approved in multiple myeloma, despite their activity having been seen in other B-cell malignancies. With the tolerability seen in the combination of IBR and DEX, combination with another oral agent is convenient. This is especially true for patients with asymptomatic progression, and for those who need an all-oral approach to avoid time off from work, to limit time spent traveling to an infusion center, to limit time spent at an infusion center, and to allow time for activities of daily living without skipping treatment. The combination of IBR/LEN/DEX was generally well tolerated with the most common grade 3/4 AEs being decreased lymphocyte count (30.9%) and anemia (15.4%). One patient had a grade 5 AE (sepsis and death) that was deemed not attributable to treatment and occurred in the setting of progressive disease. This same patient had a dose reduction after cycle 1 due to myalgia and fatigue. Otherwise, the majority of AEs were grade 1–2 and were managed with supportive care, and there were no other dose reductions.

The ORR (≥PR) of IBR/LEN/DEX was 15.4% (90% CI: 2.8–41.0%) with a CBR of 61.5% (90% CI: 35.5–83.4%).

Although disease stabilization is not defined as a response to therapy, the sustained SD in heavily pre-treated RR MM with tolerable toxicities compared to that under other combination treatments suggests that further investigation of IBR plus an IMiD, especially in patients with slow or asymptomatic disease progression, is warranted.

The study closed on 10 May 2023 before completion due to slow accrual. One of the barriers to enrollment was limiting those exposed to LEN to patients who had progressed on LEN 10 mg or less, but not on a higher dose. The COVID-19 pandemic also led to us halting accrual for up to 6 months. A number of measures were undertaken to improve accrual, including opening enrollment to patients who had undergone one prior line of therapy. Unfortunately, accrual did not substantially improve with these measures, prompting study closure and completion.

In conclusion, the combination of IBR with LEN and DEX was well tolerated, with clinical activity seen. Potential future directions for an all-oral combination with IBR and the accompanying real-world advantages include next-generation IMiDs such as pomalidomide, and the potent oral degraders in the CELMod category, specifically iberdomide and mezigdomide [9,22,23]

## Figures and Tables

**Figure 1 cancers-17-02433-f001:**
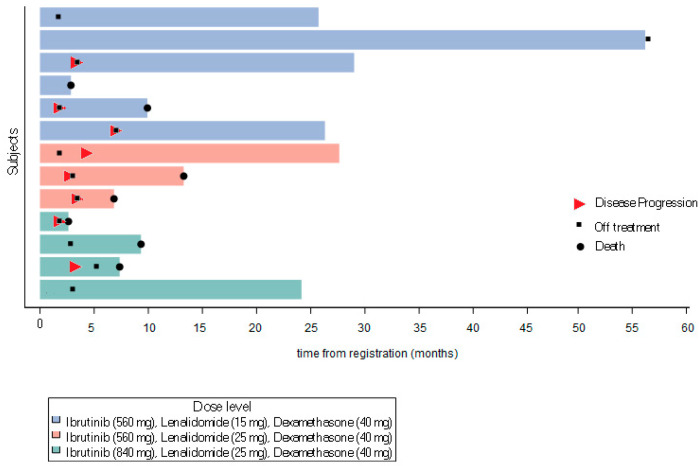
Swimmers plot.

**Figure 2 cancers-17-02433-f002:**
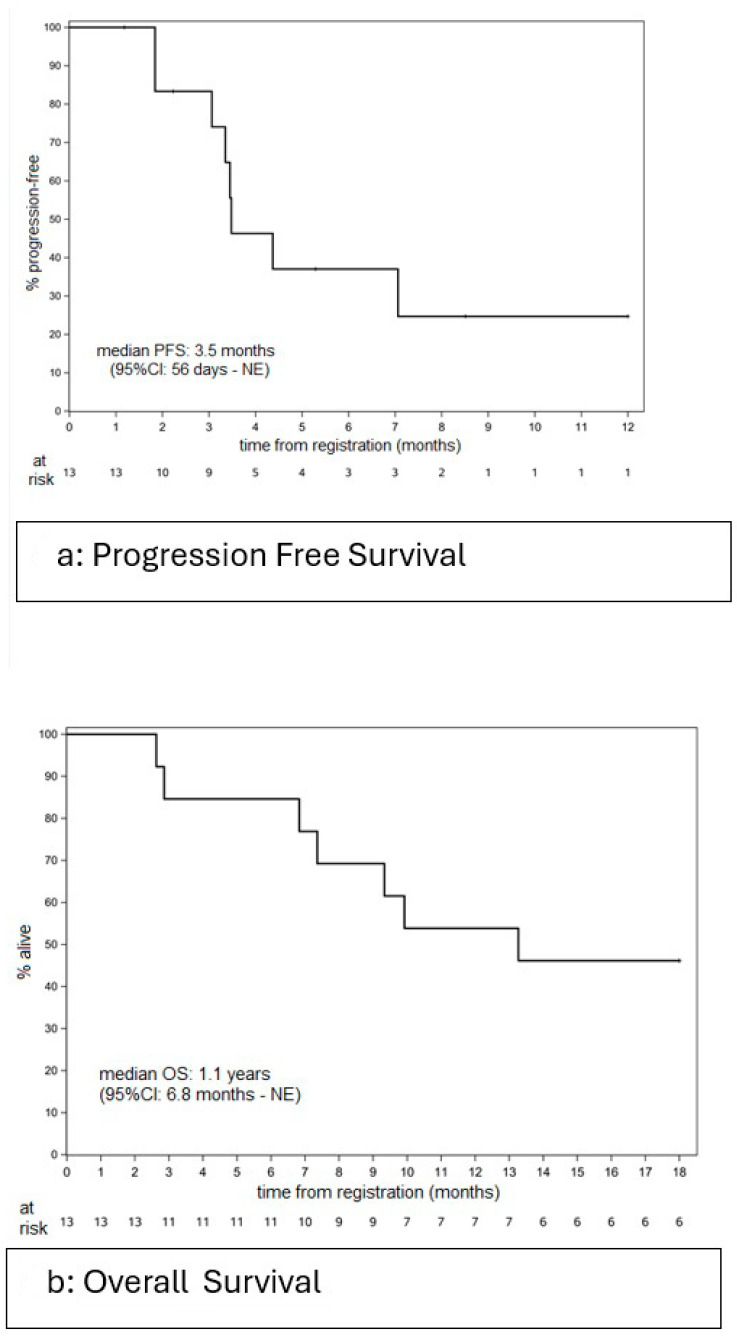
Please add the corresponding content of this part.

**Table 1 cancers-17-02433-t001:** Patient and disease characteristics at registration.

Characteristic	No. Patients (%) (n = 13)
Median age, years (range)	64 (45–84)
Sex	
Male	5 (38.5%)
Female	8 (61.5%)
Race	
Black or African American/African Heritage	1 (7.7%)
White	10 (76.9%)
Not provided	2 (15.4%)
Ethnicity	
Hispanic/Latino	1 (7.7%)
Non-Hispanic/Latino	11 (84.6%)
Not provided	1 (7.7%)
ECOG Performance Score	
0	3 (23.1%)
1	10 (76.9%)
Prior autologous transplant	10 (76.9%)
Number of lines of prior systemic therapy	
2	5 (38.5%)
3	3 (23.1%)
4–9	5 (38.5%)
Prior systemic therapy	
Bortezomib	13 (100%)
Carfilzomib	5 (38.5%)
CAR-T cell therapy	1 (7.7%)
CB-839 HCI	1 (7.7%)
Cyclophosphamide	2 (15.4%)
Daratumumab	7 (53.8%)
Iberdomide	1 (7.7%)
Ixazomib	2 (15.4%)
Lenalidomide	12 (92.3%)
Pomalidomide	4 (30.8%)
Thalidomide	1 (7.7%)
No measurable soft tissue plasmacytomas or extramedullary disease present	13 (100%)
FISH abnormalities	
No abnormal findings	1 (7.7%)
Abnormal findings including:	12 (92.3%)
del(13q) —3 pts	
del (17p)—3 pts	
t(4; 14)—1 pt	
t(11:14)—3 pts	
gain1q—4 pts	
loss1p—1 pt	
trisomy—2 pts	
MYC abnormality—1 pt	

**Table 2 cancers-17-02433-t002:** Grade 2–5 toxicities reported regardless of attribution (n = 13).

Toxicity	Grade (n = 13)			
	2	3	4	5
Abdominal pain	7.7%			
Alanine aminotransferase increase		7.7%		
Alkaline phosphatase increase	7.7%			
Anemia	7.7%	15.4%		
Anorexia	7.7%			
Aspartate aminotransferase increase		7.7%		
Back pain	23.1%			
Blood bilirubin increase		7.7%		
Cellulitis		7.7%		
Chest pain—cardiac		7.7%		
COVID-19	7.7%	7.7%		
Chronic kidney disease		7.7%		
Confusion	7.7%			
Conjunctivitis	7.7%			
Constipation	7.7%			
Creatinine increase	7.7%			
Cytopenia		7.7%		
Dehydration	7.7%			
Diarrhea	23.1%			
Dyspepsia	15.4%			
Dyspnea	7.7%			
E. coli (urine)	7.7%			
Edema (face)	7.7%			
Fatigue	30.8%	7.7%		
Febrile neutropenia		7.7%		
Gastroesophageal reflux disease	15.4%			
Generalized muscle weakness	7.7%			
Hypoalbuminemia	7.7%			
Hypocalcemia		7.7%		
Hypokalemia		7.7%		
Hypophosphatemia	7.7%	7.7%		
Hypotension	7.7%			
Insomnia	7.7%			
Lung infection		7.7%		
Lymphocyte count decrease		30.8%	7.7%	
Myalgia	7.7%			
Nail infection	7.7%			
Nasal congestion	7.7%			
Non-viral hepatitis		7.7%		
Pain in extremity	7.7%			
Platelet count decrease	15.4%		7.7%	
Pleural effusion		7.7%		
Rash, maculo-papular	15.4%			
Sepsis syndrome				7.7%
Sinus tachycardia	7.7%			
Sinusitis		7.7%		
Skin infection		7.7%		
Soft tissue infection		7.7%		
Somnolence	7.7%			
Upper respiratory infection	15.4%			
Urinary tract infection		7.7%		
Weight loss	15.4%			
White blood cell decrease	7.7%			

## Data Availability

Data stored at NCI with Alliance.

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
