# Peer review of "Ibrutinib in Combination with Lenalidomide Revlimid/Dexamethasone in Relapsed/Refractory Multiple Myeloma (AFT-15)"

_cancers, 2025, doi:10.3390/cancers17152433_

Round 1

Reviewer 1 Report

Comments and Suggestions for Authors

1) Do you have data about risk stratification (R-ISS, R2-ISS, FISH) of each patient?

2) Patients who were refractory to len10 mg had a different outcome from naive/exposed?

Author Response

we thank you so much for the comments.

1) Reviewer comment:  Do you have data about risk stratification (R-ISS, R2-ISS, FISH) of each patient?: 

Answer: these are patients with multiple lines of therapy. 61.6% have had 3-9 lines of therapy. B2M was not required as staging after several relapse does not show significant difference in response. the FISH however, is most impactful. these are the breakdown this is in Table 1.  8 of the 13 evaluable patients had high risk. 

del(13q)  - 3 pts

del (17p) - 3 pts

t(4; 14) - 1 pt

t(11:14) - 3 pts

gain1q - 4 pts

loss1p - 1 pt

trisomy - 2 pts

MYC abnormality - 1 pt

2) reviewer comment: Patients who were refractory to len10 mg had a different outcome from naive/exposed?

answer: 92.3% had been exposed and refractory to Lenalidomide as part of maintenance with their initial first line therapy.

added " All 13 patients were exposed and refractory to an IMID (namely, lenalidomide [92.3%], pomalidomide [30.8%] or thalidomide [7.7%]) and to a proteasome inhibitor (namely, Bortezomib [100%], Carfilzomib [38.5%] or Ixazomib [15.4%])

Reviewer 2 Report

Comments and Suggestions for Authors

This study presented a 3+3 phase I clinical trial of ibrutinib plus lenalidomide in R/R multiple myeloma and showed the combination of IBR with LEN and DEX was well tolerated, though the clinical benefit was not promising. Please allow me to have some suggestions.

  1. Line 307 need to be rewritten.
  2. Can the naration of how the clinical trial was designed and done be shorten, since there is protocol for 3+3 Phase I clinical trial?

Author Response

We thank reviewer for the review and Favorable comment

Reviewer comment:  Line 307 need to be rewritten

Answer:  we have done this

Reviewer comment: Can the naration of how the clinical trial was designed and done be shorten, since there is protocol for 3+3 Phase I clinical trial

Answer:  This study was done through Alliance/AFT15, and they do have a standard method of writing the design. this was the final approved version.  unfortunately we cannot change. thanks so much

Reviewer 3 Report

Comments and Suggestions for Authors

Adaptation of treatment of patients with multiple myeloma (MM) to the characteristics of the disease (biological phenotype, stage, MRD status), the nature of previous therapy and the patient's condition (age, somatic status, fragility status) allows the doctor to correctly formulate the treatment goal and accordingly select a treatment option ranging from immunotherapy to palliative care.

The need for practical hematology in an oral form of therapy is due, first of all, to the incurability of MM and the need of a significant proportion of patients for long-term stabilization of the disease, accompanied by an acceptable quality of life.

A combination of drugs with different mechanisms of action has proven its undeniable advantage over monotherapy options. Thus, the combination of a Bruton tyrosine kinase inhibitor (Ibrutinib) with an immunomodulator (Lenalidomide) deserves scientific and practical attention as an opportunity to significantly expand the range of drug combinations used to treat patients with RR MM.

An attractive feature of the work is not only the significant age range, but also the number of previous treatment regimens, including, among others, autologous HSCT and CART therapy.

It is much more difficult to evaluate the authors' conclusions about the clinical attractiveness of the studied combination. A small number of patients in accordance with the chosen regimen to achieve the stated goal. Hematological toxicity. A significant frequency of MM progression. Preferential response options outside the criteria for a complete response. A short observation period.

On the other hand, an actively or, more precisely, aggressively pre-treated group of patients with RR MM. Cases with unfavorable chromosomal aberrations, including del17p.

Perhaps it would be easier for readers to agree with the authors if there was information about the expanded biological phenotype of the disease with the provision of information on molecular damage associated with resistance to the drugs used. Then it would be more reasonable to discuss the effectiveness of the studied scheme for cases of multidrug-resistant RR MM.

In general, the data presented in the article solve the problems formulated by the authors. However, the conclusion about the acceptable effectiveness of the studied scheme seems premature.

Author Response

We thank the reviewer for the Favorable Comment:

Reviewer Comment: Adaptation of treatment of patients with multiple myeloma (MM) to the characteristics of the disease (biological phenotype, stage, MRD status), the nature of previous therapy and the patient's condition (age, somatic status, fragility status) allows the doctor to correctly formulate the treatment goal and accordingly select a treatment option ranging from immunotherapy to palliative care.

The need for practical hematology in an oral form of therapy is due, first of all, to the incurability of MM and the need of a significant proportion of patients for long-term stabilization of the disease, accompanied by an acceptable quality of life.

A combination of drugs with different mechanisms of action has proven its undeniable advantage over monotherapy options. Thus, the combination of a Bruton tyrosine kinase inhibitor (Ibrutinib) with an immunomodulator (Lenalidomide) deserves scientific and practical attention as an opportunity to significantly expand the range of drug combinations used to treat patients with RR MM.

An attractive feature of the work is not only the significant age range, but also the number of previous treatment regimens, including, among others, autologous HSCT and CART therapy.

It is much more difficult to evaluate the authors' conclusions about the clinical attractiveness of the studied combination. A small number of patients in accordance with the chosen regimen to achieve the stated goal. Hematological toxicity. A significant frequency of MM progression. Preferential response options outside the criteria for a complete response. A short observation period.

On the other hand, an actively or, more precisely, aggressively pre-treated group of patients with RR MM. Cases with unfavorable chromosomal aberrations, including del17p.

Perhaps it would be easier for readers to agree with the authors if there was information about the expanded biological phenotype of the disease with the provision of information on molecular damage associated with resistance to the drugs used. Then it would be more reasonable to discuss the effectiveness of the studied scheme for cases of multidrug-resistant RR MM.

In general, the data presented in the article solve the problems formulated by the authors. However, the conclusion about the acceptable effectiveness of the studied scheme seems premature.

Answer: We thank the reviewer for the comments.  we agree that the data is small but valuable information and tool for all future all oral agents, especially combination of Ibrutinib with the celmods.

Reviewer 4 Report

Comments and Suggestions for Authors

The authors describe their phase I trial of ibrutinib + lenalidomide, which could have been an interesting combination based on (limited) preclinical data. Unfortunately, responses are disappointing and in the rapidly evolving myeloma treatment landscape with various more potent options, the execution of the suggested follow-up trials with the combination of ibrutinib + another IMiD/CELMod will probably not have the highest priority. Toxicity was however acceptabel in this phase I trial with a heavily pretreated population.

Of the treated patients, 12/13 have been treated with lenalidomide. It would be interesting to know if all of these patients received len maintenance until disease progression and should therefore be considered len refractory. 

Despite the high doses of ibrutinib administered in this trial, well-known toxicities like atrial fibrillation and haemorrhage are not described. Do the authors think this is mainly a results of the small population size and limited treatment duration in most patients, or does it reflect a difference in toxicity profile in different diseases?

The authors describe several reasons why accrual was slow and the study ultimately had to close before the planned accrual was reached. Do you think also competing (study) treatment options with for example bispecific antibodies and/or CAR-T cell therapies may have played a role here? 

Author Response

We thank the reviewer for the comments

Comment: The authors describe their phase I trial of ibrutinib + lenalidomide, which could have been an interesting combination based on (limited) preclinical data. Unfortunately, responses are disappointing and in the rapidly evolving myeloma treatment landscape with various more potent options, the execution of the suggested follow-up trials with the combination of ibrutinib + another IMiD/CELMod will probably not have the highest priority. Toxicity was however acceptabel in this phase I trial with a heavily pretreated population.

Of the treated patients, 12/13 have been treated with lenalidomide. It would be interesting to know if all of these patients received len maintenance until disease progression and should therefore be considered len refractory. 

Answer:  yes all these patients were refractory to Lenalidomide 10 mg or less. patients who were on higher than 10 mg were excluded from the study. we have amended the manuscript to read" .  All 13 patients were exposed and refractory to an IMID (namely, lenalidomide [92.3%], pomalidomide [30.8%] or thalidomide [7.7%]) and to a proteasome inhibitor (namely, Bortezomib [100%], Carfilzomib [38.5%] or Ixazomib [15.4%]).

Comment: Despite the high doses of ibrutinib administered in this trial, well-known toxicities like atrial fibrillation and haemorrhage are not described. Do the authors think this is mainly a results of the small population size and limited treatment duration in most patients, or does it reflect a difference in toxicity profile in different diseases?

Answer: it is very likely that the biology of Myeloma, being different than lymphoma and CLL, Myeloma patients seem to tolerate well. one patient stayed on the combination for over 50 months.  also agree the small Number of patients does not give a detailed info 

Comment: The authors describe several reasons why accrual was slow and the study ultimately had to close before the planned accrual was reached. Do you think also competing (study) treatment options with for example bispecific antibodies and/or CAR-T cell therapies may have played a role here? 

Answer:  CAR-T was available, Bispecific not yet at the time of study conduct until the last year. both definitely may have played a role.  also many patients were on higher  than 10 mg dose of lenalidomide at time of progression so were excluded. this was a major reason. a combination of Ibrutinib and Pomalidomide may have accrued  better.  the tolerability though with Ibrutinib and an imid is worth noted for future studies.   enough for their patients who was progressing